# Investigating the nature of open science practices across complementary, alternative, and integrative medicine journals: An audit

Jeremy Y. Ng [1,2,3]*, Brenda Lin[1,2], Tisha Parikh[1,2], Holger Cramer[1,2], David Moher[3,4]

**1** Institute of General Practice and Interprofessional Care, University Hospital Tübingen, Tübingen, Germany, **2** Robert Bosch Center for Integrative Medicine and Health, Bosch Health Campus, Stuttgart, Germany, **3** Ottawa Hospital Research Institute, Centre for Journalology, Ottawa Methods Centre, Ottawa, Ontario, Canada, **4** School of Epidemiology and Public Health, University of Ottawa, Ottawa, Canada

* ngjy2@mcmaster.ca, jeremy.ng@med.uni-tuebingen.de

**Data Availability Statement:** All relevant data are included in this manuscript or posted on Open Science Framework: https://doi.org/10.17605/OSF.IO/S7G6P.

## Abstract

### Background

Open science practices are implemented across many scientific fields to improve transparency and reproducibility in research. Complementary, alternative, and integrative medicine (CAIM) is a growing field that may benefit from adoption of open science practices. The efficacy and safety of CAIM practices, a popular concern with the field, can be validated or refuted through transparent and reliable research. Investigating open science practices across CAIM journals by using the Transparency and Openness Promotion (TOP) guidelines can potentially promote open science practices across CAIM journals. The purpose of this study is to conduct an audit that compares and ranks open science practices adopted by CAIM journals against TOP guidelines laid out by the Center for Open Science (COS).

### Methods

CAIM-specific journals with titles containing the words "complementary", "alternative" and/or "integrative" were included in this audit. Each of the eight TOP criteria were used to extract open science practices from each of the CAIM journals. Data was summarized by the TOP guideline and ranked using the TOP Factor to identify commonalities and differences in practices across the included journals.

### Results

A total of 19 CAIM journals were included in this audit. Across all journals, the mean TOP Factor was 2.95 with a median score of 2. The findings of this study reveal high variability among the open science practices required by journals in this field. Four journals (21%) had a final TOP score of 0, while the total scores of the remaining 15 (79%) ranged from 1 to 8.

### Conclusion

While several studies have audited open science practices across discipline-specific journals, none have focused on CAIM journals. The results of this study indicate that CAIM

**Funding:** The author(s) received no specific funding for this work.

**Competing interests:** The authors have declared that no competing interests exist.

**Abbreviations:** CAIM, complementary, alternative, and integrative medicine; COS, Center for Open Science; TOP, Transparency and Openness Promotion; ASJC, All Science Journal Classification.

journals provide minimal guidelines to encourage or require authors to adhere to open science practices and there is an opportunity to improve the use of open science practices in the field.

## Background

Open science is an emerging movement aimed at making scientific research and data more transparent and accessible. Researchers and those in the publishing community promote collaboration and reproducibility in research by implementing open science practices, which can include: open data, referring to readily available study data; open access, referring to accessible distribution or publication of data; and transparent peer review processes [1–5]. Issues with certain existing research processes, including limited access to research data, and low reproducibility, have led to the growing popularity of open science [6]. One of the leading organizations advocating for open science practices in research is the Center for Open Science (COS), a non-profit organization with a mission to increase transparency, integrity, and reproducibility of research. The COS created the 2015 Transparency and Openness Promotion (TOP) guidelines published in *Science* [7] as an incentive for researchers, journal funding models, and infrastructures to integrate open science practices into their scholarly work [8]. These guidelines comprise the TOP Factor and include open science standards such as data transparency and study preregistration [7]. The TOP Factor is a metric that quantitatively measures the extent to which the journals have adopted, require, and encourage the TOP guidelines in their publications [9]. The usage of open science practices provides many benefits for scholarly journals. For example, the public availability of study data allows for more transparent science and encourages the reuse of existing data to avoid unnecessary redundancy within the scholarly literature [10–12]. Nonetheless, there are still many barriers to implementing open science. Reduced scientific flexibility brought about by a higher focus on confirmatory rather than exploratory research, increased time commitments with preregistration, quality control, and peer review, and increased publishing costs may disincentivize researchers and journals from adopting open science practices [11,13,14]. However, the ability of open science to make the field more reliable and accessible, should drive journals to encourage and require open science practices as part of their publishing processes and, in turn, promote researchers to conduct research in line with these practices [15].

Open science practices specific to journals have been audited in various other fields of research, such as in psychology [16], the communication sciences [17] and the medical sciences [18]. The implementation of open science practices in complementary, alternative, and integrative medicine (CAIM) journals has not yet been investigated. CAIM is a field of medicine involving healthcare practices that are not generally used in conventional medical care [19]. Complementary therapies refer to those used in conjunction with conventional medicine, whereas alternative therapies are those used in place of conventional medicine [20]. More recently, integrative medicine has become an increasingly popular field which uses both conventional and complementary approaches to medicine to provide a more holistic form of care [19–21]. For the purpose of this study, all the terms described will be collectively referred to as CAIM. Though many CAIM therapies have been used for centuries, evidence-based practices are still relatively new to this field which stems from being a neglected area of original research that is performed by the relatively small subset of CAIM clinical practitioners [22–24]. Though some areas of CAIM (e.g., mindfulness and meditation research) [25,26] have a large evidence

basis, the paucity of many areas of CAIM research is due to the lack of funding and financial incentive [22,27], inadequate research training of many CAIM practitioners and their uncertainty towards the fundamental tenets of the scientific process [24,28,29], and stigma from biomedical researchers around the supposed biological implausibility of CAIM therapies. Even when clinical evidence supports the biological plausibility of CAIM therapies, its efficacy is still often questioned [27]. Regardless, CAIM therapies remain a popular option among patients which justifies the need for improved research quality within this field [23,28,29]. A growing number of CAIM publications over the last several decades further highlights an increased interest in the field by researchers and practitioners [30–32]. The adoption of open science practices in CAIM can provide a greater availability of safety and efficacy profiles to deliver a higher quality of care for patients [33]. To look for trends in open science practices in CAIM journals, a journal audit can be conducted to investigate the degree of open science requirement in a journal's manuscript submission guidelines. Thus, the purpose of this study is to conduct an audit that compares and ranks open science practices adopted by CAIM journals against TOP guidelines laid out by the Center for Open Science (COS).

## Methods

### Approach and open science statement

CAIM journals were identified and items from the TOP guidelines were used to extract data from each journal [34]. These items were assessed using the TOP rubric [35] to find the TOP Factor and determine which open science practices are implemented, encouraged, and required for their publications. The protocol, study material, and blank extraction form, were made available on Open Science Framework (OSF) [36].

### Journal selection

A list of CAIM journals was obtained from Table 2 of Ng's (2021) bibliometric analysis of CAIM journals [37]. Ng conducted a search on Scopus for CAIM journals. Compared to other databases such as Web of Science, Scopus includes more CAIM-categorized journals in its database [38]. The search was based on journals belonging to the category of "Complementary and Alternative Medicine" (Code 2707), of the All-Science Journal Classification (ASJC) [37].

### Eligibility criteria

Literature on any CAIM topic can be found in a general CAIM journal, but general CAIM research is not likely to be published in journals on specific topics (e.g. homeopathy). Hence, the list obtained from Table 2 of Ng's bibliometric analysis was modified to include only journals with the exact words "complementary", "alternative", and/or "integrative" in the journal title [37]. This step ensured the audit only included general CAIM journals rather than journals on specific topics in CAIM. The modified list was further narrowed to exclude journals that were discontinued, renamed, inaccessible via their website, incorporated into other journals already found in the list, or only published in print. In other words, the revised list only includes journals that are currently active and accessible at the time of data collection. **Table 1** contains the comprehensive list of journals along with the exclusion criteria that were applied.

### TOP factor

The TOP guidelines are a set of eight standards (**Table 2**) that journals are encouraged to adopt to enhance open science practices [7]. The TOP Factor is a metric that rates how well journals implement each of the TOP guidelines [9]. The TOP Factor is scored using the TOP rubric, a publicly

**Table 1. List of journals containing the words "complementary", "alternative" and/or "integrative" from Ng et al.'s bibliometric analysis, including their active status [37].**

| Source Title | Publisher | ISSN | Status* |
|---|---|---|---|
| Advances in Integrative Medicine | Elsevier | 2212–9588 | Active |
| African Journal of Traditional, Complementary and Alternative Medicines | African Networks on Ethnomedicines | 0189–6016 | Inactive |
| Alternative and Complementary Therapies | Mary Ann Liebert | 1076–2809 | Renamed to "Integrative and Complementary Therapies" |
| Alternative Medicine Alert | Future Medicine Ltd. | 1081–4000 | Inactive |
| Alternative Medicine Review | Thorne Reasearch Inc. | 1089–5159 | Inactive |
| Alternative Therapies in Health and Medicine | InnoVision Communications | 1078–6791 | Active |
| Alternative Therapies in Womens Health | American Health Consultant | 1522–3396 | Inactive |
| BMC Complementary and Alternative Medicine | Springer Nature | 1472–6882 | Renamed to "BMC Complementary Medicine and Therapies" |
| BMC Complementary Medicine and Therapies | Springer Nature | 2662–7671 | Active |
| Chinese Journal of Integrative Medicine | Springer Nature | 1672–0415 | Active |
| Complementary Health Practice Review | SAGE | 1533–2101 | Inactive |
| Complementary Medicine Research | Karger | 2504–2092 | Active |
| Complementary Therapies in Clinical Practice | Elsevier | 1744–3881 | Active |
| Complementary Therapies in Medicine | Elsevier | 0965–2299 | Active |
| Complementary Therapies in Nursing and Midwifery | Elsevier | 1353–6117 | Inactive |
| European Journal of Integrative Medicine | Elsevier | 1876–3820 | Active |
| Evidence-based Complementary and Alternative Medicine | Hindawi | 1741-427X | Active |
| Evidence-Based Integrative Medicine | Springer Nature | 1176–2330 | Inactive |
| Focus on Alternative and Complementary Therapies | Wiley-Blackwell | 1465–3753 | Inactive |
| Integrative and Complementary Therapies | Mary Ann Liebert | 2768–3192 | Active |
| Integrative Cancer Therapies | SAGE | 1534–7354 | Active |
| Integrative Medicine | InnoVision Communications | 1546-993X | Active |
| Integrative Medicine Alert | American Health Consultants, Inc. | 2325–2812 | Inactive |
| Integrative Medicine Insights | Libertas Academica | 1177–3936 | Inactive |
| Integrative Medicine Research | Elsevier | 2213–4220 | Active |
| Journal of Alternative and Complementary Medicine | Mary Ann Liebert | 1075–5535 | Renamed to "Journal of Integrative and Complementary Medicine" |
| Journal of Integrative and Complementary Medicine | Mary Ann Liebert | 2768–3613 | Active |

*(Continued)*

**Table 1.** (Continued)

| Source Title | Publisher | ISSN | Status* |
|---|---|---|---|
| Journal of Ayurveda and Integrative Medicine | Elsevier | 0975–9476 | Active |
| Journal of Complementary and Integrative Medicine | Walter de Gruyter | 1553–3840 | Active |
| Journal of Complementary Medicine | Australian Pharmaceutical Publishing Co., Ltd. | 1446–8263 | Inactive |
| Journal of Evidence-Based Complementary and Alternative Medicine | SAGE | 2156–5872 | Renamed to "Journal of Evidence-Based Integrative Medicine" |
| Journal of Evidence-Based Integrative Medicine | SAGE | 2515-690X | Active |
| Journal of Experimental and Integrative Medicine | Gesdav | 1309–4572 | Inactive |
| Journal of Integrative Medicine | Elsevier | 2095–4964 | Active |
| Journal of the Society for Integrative Oncology | B.C. Decker Inc. | 1715-894X | Inactive |
| Journal of Traditional and Complementary Medicine | Elsevier | 2225–4110 | Active |
| Scientific Review of Alternative Medicine | Prometheus Books Inc. | 1095–0656 | Inactive |
| Seminars in Preventive and Alternative Medicine | Elsevier | 1556–4061 | Inactive |
| Traditional and Integrative Medicine | Tehran University of Medical Sciences | 2476–5104 | Active |

*Data extractions and TOP scoring will only be conducted on journals with an "Active" status.

available rubric created by the COS to guide journal assessment [35]. The TOP rubric includes four possible levels where journals can be ranked, from 0 to 3 [35]. Each TOP guideline has individual requirements for each score, with a higher score indicating stronger adherence to each guideline. For example, for data sharing, a score of 0 indicates that the journal only encourages or does not mention data sharing. A score of 1 indicates that the journal requires the author to disclose where data are publicly accessible. A score of 2 indicates that the journal mandates that authors share their data except for certain circumstances such as sensitive health data and proprietary data. A score of 3 implies that the journal not only requires data sharing, but also includes a verification process to ensure that the data is consistent with the reported findings in the published article [35]. The highest possible TOP Factor a journal can receive is 24.

## Data extraction and assessment of journal practices

The first draft of the data extraction form was developed by BL and TP, which was reviewed by JYN with careful detail. The revised data extraction form was then circulated to HC and DM for their feedback and amended before beginning data extraction. The following items were extracted from the journals being evaluated in this study: website Uniform Resource Locator (URL), International Standard Serial Number (ISSN), Journal Citation Reports (JCR) impact factor, publisher, whether the journal is available in print or online, first year of print publication (if applicable), first year of online publication, and the specific text associated with the qualitative features as outlined by the items in the TOP guidelines.

The degree to which open science practices were required by journals was assessed using the TOP Factor [39]. Journals can be submitted for evaluation by the COS or journals can be self-

**Table 2. Summary of the eight transparency and openness promotion (TOP) guidelines [8].**

| TOP Guideline | Definition Summary |
|---|---|
| Data Citation | "Citation of articles is routine and well-formulated. Similar standards can be applied to citation of data, code, and materials to recognize and credit these as original intellectual contributions." |
| Data Transparency | "Transparency guidelines for data, analytic methods, and research materials are conceptually distinct. They are presented together as the process principles are similar for each. However, a journal could adopt different levels for each with minor modifications of the templates." |
| Analytical Code Transparency | |
| Materials Transparency | |
| Design & Analysis Transparency (Reporting Guidelines) | "Standards for reporting research design and analysis should maximize transparency about the research process and minimize potential for vague or incomplete reporting of the methodology. The standards for data, analytic methods, and research materials above provide general guidelines for making such material available." |
| Study Preregistration | "Preregistration of studies is a means of making research more discoverable even if it does not get published. By encouraging or requiring preregistration, journals increase the likelihood of discoverability of research that is not ultimately published." |
| Analysis Plan Preregistration | "Preregistration of Analysis Plans certifies the distinction between confirmatory and exploratory research. Preregistration of Analysis Plans supercedes Study Preregistration above. If a transparency standard for analysis plans is adopted, then the text below is adopted instead of text in Preregistration of Studies. An exception to this rule is if a stronger transparency standard is adopted for studies than for analysis plans. In that case, minor edits of the text below may be needed to avoid competing language with above." |
| Replication | "The transparency standards above account for reproducibility of the reported results based on the originating data, and for sharing sufficient information to conduct an independent replication. While not formally a transparency standard for authors, this section addresses journal guidelines for consideration of independent replications for publication." |

evaluated using the TOP rubric and submitted to the COS for verification [34]. We self-assessed journals and submitted our evaluations to the COS through a Google form posted on their website [40]. Journals that were found to have already been assessed by the COS were used to pilot test the data extraction procedure [35]. A pilot data extraction procedure was performed by BL and TP on 5 journals in the list, of which 2 were previously assessed by COS. The purpose of the pilot extraction was to allow an opportunity for standardization among scorers before proceeding to the independent extractions. Most of the journals selected for the pilot had different publishers to account for anticipated differences in the requirement of open science practices. The extracted data was compared between BL and TP, then carefully reviewed by JYN. The remaining conflicting TOP ratings were resolved with consultation from HC and DM.

## Data analysis and presentation

The CAIM journals' open science practices were evaluated using the TOP rubric to identify the individual scores for each journal. CAIM journals were then ranked by their overall TOP Factor scores. To evaluate which TOP guidelines were most engaged with by the journals, scores for each TOP guideline were summed across all journals to find the TOP Factor.

## Results

### Description of journals

A total of 19 CAIM journals were included in this audit, two of which were previously assessed by the COS. A flow chart of the journal exclusion process can be found in **Fig 1**. **Table 3**

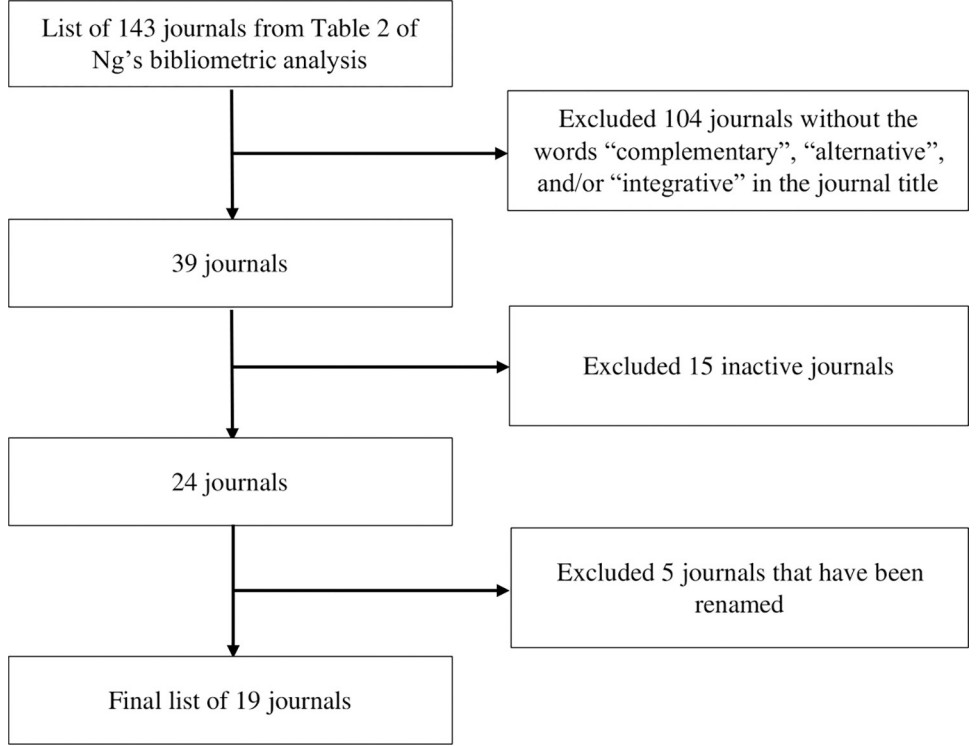

**Fig 1. Flowchart of exclusion criteria for journals from Table 2 of Ng et al.'s bibliometric analysis.**

includes each journal's URL and general information about their impact factors, whether they are available in print or online, and their first year of publication. In summary, the journals' impact factors ranged from 1.449 to 4.473. Seventeen journals (89%) are available in print and online and two (11%) are only available online. The first year of online publication ranges from 1995 to 2022.

## Overall TOP scores

**Table 4** shows the TOP guideline scores and TOP Factor calculated for all 19 journals. The mean TOP Factor across all journals was 2.95, with a median score of 2 and standard deviation of 2.63. The journals with the highest overall TOP Factors were *Evidence-based Complementary and Alternative Medicine* (8), *Complementary Medicine Research* (7), *Integrative Cancer Therapies* (7), and *Journal of Evidence-Based Integrative Medicine* (7). Four journals (21%) had a TOP Factor of 0, including *African Journal of Traditional, Complementary and Alternative Medicines*, *Chinese Journal of Integrative Medicine*, *Integrative Medicine*, and *Journal of Complementary and Integrative Medicine*. Information about the specific text associated with each TOP guideline for each journal can be found in **S1 Table** or on OSF here: https://osf.io/jmkyn.

## Data citation

Six journals (32%) provided no information regarding data citation practices and received a score of 0 in this category. Ten journals (53%) encouraged citation of datasets, though the practice was not required nor mandated for publication, receiving a score of 1. As noted in

**Table 3. General information about the 19 CAIM journals.**

| | Journal URL | Journal Impact Factor | Print or Online Journal | First Year of Print Publication (If Applicable) | First Year of Online Publication |
|---|---|---|---|---|---|
| *Advances in Integrative Medicine* | https://www.sciencedirect.com/journal/advances-in-integrative-medicine | N/A | Both | 2014 | 2014 |
| *African Journal of Traditional, Complementary and Alternative Medicines* | https://www.ajol.info/index.php/ajtcam | 0.553 (2015) | Both | 2004 | 2006 |
| *Alternative Therapies in Health and Medicine* | http://www.alternative-therapies.com | N/A | Both | 1995 | 1995 |
| *BMC Complementary Medicine and Therapies* | https://bmccomplementmedtherapies.biomedcentral.com | 2.838 | Online | N/A | 2001 |
| *Chinese Journal of Integrative Medicine* | https://www.springer.com/journal/11655 | N/A | Both | 2003 | 2003 |
| *Complementary Medicine Research* | https://www.karger.com/cmr | 1.449 | Both | 1994 | 2000 |
| *Complementary Therapies in Clinical Practice* | https://www.sciencedirect.com/journal/complementary-therapies-in-clinical-practice | 3.577 | Both | 2005 | 2005 |
| *Complementary Therapies in Medicine* | https://www.sciencedirect.com/journal/complementary-therapies-in-medicine | 3.335 | Both | 1993 | 2006 |
| *European Journal of Integrative Medicine* | https://www.sciencedirect.com/journal/european-journal-of-integrative-medicine | 1.813 | Both | 2008 | 2008 |
| *Evidence-based Complementary and Alternative Medicine* | https://www.hindawi.com/journals/ecam/ | 2.65 | Both | 2004 | 2004 |
| *Integrative Cancer Therapies* | https://journals.sagepub.com/home/ict | 3.077 | Both | 2002 | 2005 |
| *Integrative Medicine* | http://www.imjournal.com/index.cfm | n/a | Both | 2002 | 2002 |
| *Integrative Medicine Research* | https://www.sciencedirect.com/journal/integrative-medicine-research | 4.473 | Both | 2012 | 2012 |
| *Journal of Ayurveda and Integrative Medicine* | https://www.sciencedirect.com/journal/journal-of-ayurveda-and-integrative-medicine | N/A | Both | 2010 | 2010 |
| *Journal of Complementary and Integrative Medicine* | https://www.degruyter.com/journal/key/jcim/html?lang=en | N/A | Online | N/A | 2004 |
| *Journal of Evidence-Based Integrative Medicine* | https://journals.sagepub.com/home/chp | N/A | Both | 1995 | 2018 |
| *Journal of Integrative and Complementary Medicine* | https://www.liebertpub.com/loi/acm | 2.381 | Both | 2022 | 2022 |
| *Journal of Integrative Medicine* | https://www.sciencedirect.com/journal/journal-of-integrative-medicine | 3.951 | Both | 2013 | 2013 |
| *Journal of Traditional and Complementary Medicine* | https://www.sciencedirect.com/journal/journal-of-traditional-and-complementary-medicine | 4.221 | Both | 2011 | 2011 |
| *Traditional and Integrative Medicine* | https://jtim.tums.ac.ir | N/A | Both | 2016 | 2016 |

the TOP rubric, the word "should" was presumed as encouragement rather than a requirement [35]. Three journals (16%) required appropriate citation for all data obtained elsewhere, resulting in a score of 2. No journals requested enforced data citation as a condition for publication.

## Data, analytical code, and research materials transparency

The majority of journals (68%) received scores of 0 for the data transparency, analytical code transparency, and research materials transparency categories. Of these, 8 journals (42%) had no mention of data sharing, while 5 (26%) only encouraged it.

**Table 4. TOP guideline scores and TOP factor for the 19 CAIM journals.**

| | Data Citation | Data Transparency | Analytical Code Transparency | Materials Transparency | Reporting Guidelines | Study Preregistration | Analysis Plan Preregistration | Replication | TOP Factor |
|---|---|---|---|---|---|---|---|---|---|
| *Evidence-based Complementary and Alternative Medicine* | 1 | 2 | 1 | 1 | 1 | 1 | 0 | 1 | 8 |
| *Complementary Medicine Research* | 1 | 1 | 1 | 1 | 2 | 0 | 1 | 0 | 7 |
| *Integrative Cancer Therapies* | 2 | 3 | 0 | 0 | 2 | 0 | 0 | 0 | 7 |
| *Journal of Evidence-Based Integrative Medicine\** | 2 | 3 | 0 | 0 | 2 | 0 | 0 | 0 | 7 |
| *BMC Complementary Medicine and Therapies\*†* | 2 | 1 | 1 | 0 | 1 | 0 | 0 | 0 | 5 |
| *Integrative Medicine Research* | 1 | 2 | 0 | 0 | 2 | 0 | 0 | 0 | 4 |
| *Advances in Integrative Medicine* | 1 | 0 | 0 | 0 | 2 | 0 | 0 | 0 | 3 |
| *European Journal of Integrative Medicine* | 1 | 0 | 0 | 0 | 1 | 1 | 0 | 0 | 3 |
| *Complementary Therapies in Clinical Practice* | 1 | 0 | 0 | 0 | 0 | 1 | 0 | 0 | 2 |
| *Complementary Therapies in Medicine\*†* | 1 | 0 | 0 | 0 | 1 | 0 | 0 | 0 | 2 |
| *Journal of Ayurveda and Integrative Medicine\** | 0 | 0 | 0 | 0 | 2 | 0 | 0 | 0 | 2 |
| *Journal of Integrative and Complementary Medicine* | 1 | 0 | 0 | 0 | 1 | 0 | 0 | 0 | 2 |
| *Alternative Therapies in Health and Medicine\** | 0 | 0 | 0 | 0 | 1 | 0 | 0 | 0 | 1 |
| *Journal of Integrative Medicine* | 1 | 0 | 0 | 0 | 0 | 0 | 0 | 0 | 1 |
| *Journal of Traditional and Complementary Medicine* | 1 | 0 | 0 | 0 | 0 | 0 | 0 | 0 | 1 |
| *Traditional and Integrative Medicine* | 0 | 0 | 0 | 0 | 1 | 0 | 0 | 0 | 1 |
| *Chinese Journal of Integrative Medicine* | 0 | 0 | 0 | 0 | 0 | 0 | 0 | 0 | 0 |
| *Integrative Medicine* | 0 | 0 | 0 | 0 | 0 | 0 | 0 | 0 | 0 |
| *Journal of Complementary and Integrative Medicine* | 0 | 0 | 0 | 0 | 0 | 0 | 0 | 0 | 0 |

*Journals included in our pilot extraction.

†Journals previously assessed by the COS.

Two journals, *BMC Complementary Medicine and Therapies* and *Complementary Medicine Research* (11%) required a Data Availability Statement outlining whether and where research, including code data, can be found. Another journal, *Evidence-based Complementary and Alternative Medicine*, required that "research published in the journal must be: as reproducible as possible–sharing underlying data, code, and supporting materials wherever able." [41] All three of these journals (16%) were ranked a score of 1 in both the data transparency and analytical code transparency categories. However, *BMC Complementary Medicine and Therapies* only encouraged rather than required sharing research materials upon request, which resulted in a score of 0 for research materials transparency, unlike the other two journals which received a score of 1.

One journal, *Integrative Medicine Research*, received a score of 2 in the data transparency category for mandating a Data Availability Statement and, where applicable, noting reasons why data may not be shared. This journal, however, only encouraged code and material sharing, receiving a score of 0 for both the analytical code transparency and research materials transparency categories.

Two journals, *Integrative Cancer Therapies* and *Journal of Evidence-Based Integrative Medicine*, (11%) required publicly available data as a condition of publication, receiving a full score of 3 for data transparency. Both of these journals did not include any specific information about code or material sharing, receiving a score of 0 for these latter categories.

## Design & analysis transparency

Seven journals (37%) received a score of 0 in this category as four journals (21%) did not mention reporting guidelines at all and three journals (16%) only mentioned CONSORT, which does not encompass the majority of articles published within journals [35]. Seven journals (37%) received a score of 1 for encouraging use of reporting guidelines across several study types. As noted in the TOP rubric, the word "expect" was presumed as a requirement [35] Five journals (26%) required adherence to appropriate reporting guidelines for publication and received a score of 2. None of the journals enforced this adherence and hence were not allotted a full score of 3.

## Study and analysis plan preregistration

Sixteen journals (84%) received a score of 0 for study preregistration. Preregistration, as defined by OSF, refers to the creation and storage of the research plan in a public repository at the start of the study [42]. Of these journals, five (26%) did not mention preregistration at all and 11 (58%) only mentioned information about preregistration for clinical trials. Three journals, *Evidence-based Complementary and Alternative Medicine*, *European Journal of Integrative Medicine*, and *Complementary Therapies in Clinical Practice* (16%), received a score of 1 for including a statement about article preregistration in their authorship guidelines.

Eighteen journals (95%) received a score of 0 for analysis plan preregistration due to no explicit mention of preregistration with an analysis plan. One journal, *Complementary Medicine Research*, (5%) received a score of 1 for including a statement about analysis plan preregistration in its authorship guidelines.

## Replication

Most journals (95%) provided no information regarding submission of replication studies. One journal, *Evidence-based Complementary and Alternative Medicine*, (5%) encouraged replication studies, resulting in a score of 1 for this category.

## Discussion

### Summary of results

The purpose of this study was to conduct an audit which investigates the nature of open science practices across CAIM journals. The mean TOP Factor across all journals was found to be 2.95, with a median score of 2 and standard deviation of 2.63. The range of TOP scores reflects the variability in open science practices within CAIM journals. Four journals (21%) had a final TOP score of 0, indicating no adherence to open science practices. The highest TOP factor was an 8, out of a maximum possible of 24. Across the audit, journals most frequently adhered to some requirements of data citation and design and analysis transparency, as evidenced by the fewest scores of zero in these categories. Journals had the lowest scores in the analysis plan preregistration (this is not a common termed used in medicine. Registration is a more established and used term) and replication categories. A common trend across several categories was encouragement rather than requirement or enforcement of open science practices. TOP scores of 0 and 1, corresponding to no mention or encouragement, were more common in all categories than scores of 2 and 3, corresponding to a mandate or condition of publication. Overall, our findings suggest that CAIM journals provide minimal guidelines to encourage or require authors to adhere to open science practices, as reflected by the average TOP Factor of 2.95.

This low usage of open science practices across journals is similar to findings in other disciplines and suggests that journals have an opportunity to improve research practices through regulating the use of open science practices. For example, research shows open science practices are not highly encouraged nor required by many communication sciences and disorders journals, with a mean TOP Factor of 1.4 [17]. A similar finding was seen in an audit of open science practices across health and medical science journals, where transparency and open science practices in these journals were not frequently encouraged or mandated [18]. However, the mean TOP Factor was found to be 7, which is higher than the mean for the CAIM journals included in the present study. Similar audits were conducted on pain journals and sleep and chronobiology journals, which found that there was relatively low journal engagement with open science standards in these fields with a median TOP Factors of 3.5 and 3, respectively [43,44]. Furthermore, an audit done on clinical psychology journals found that many journal recommendations, such as use of reporting guidelines, are not frequently enforced in journals [16]. In the field of CAIM, open science practices have the potential to increase the quality reproducibility of research, as better data transparency can promote replicability and peer review to ensure reliability and credibility of findings. Open science practices in CAIM may also expedite the research process through emphasis on preregistration, for example. With improved article quality, stronger experimental evidence, and greater dissemination of CAIM research, negative perceptions of the field may be addressed. Increased public and provider trust in the field may then allow more funding and resources to be allocated for CAIM research, allowing continual improvement of research quality in the field.

With a growing interest in CAIM from patients and practitioners, evident by the numerous patient trials each year, it is becoming increasingly important to ensure that the research quality within this field of medicine is improved and standardized [45]. As many patients use research to make decisions about their healthcare, it is critical for journals to hold submitted articles to a greater standard of open science practices to avoid patient misinformation and improve provider attitudes [46,47]. For example, a study found that many patients resorted to and trusted information on CAIM therapy from medical journals, in the form of clinical trials, compared to social networks or other online media [46]. Another study also noted patients used medical and lay publications in addition to the Internet for better understanding their

CAIM options [48]. Additionally, greater journal implementation of open science practices can allow for an easier assessment of comprehensive and transparent reporting, particularly the methods and results. This likely provides increased credibility of CAIM evaluations, which is important to health care providers and patients. Various parts of the research lifecycle can also be made more accessible, allowing researchers to better build upon existing research data. While this audit focused on general CAIM journals, future research may investigate open science practices across journals on specific CAIM topics (e.g. homeopathy) to understand how research is conducted and presented between and across different CAIMs.

Given the potential benefits of open science practices for CAIM research, it is important to consider barriers to its usage and ways to overcome them. A potential reason for low TOP scores includes a low incentive for journals to implement the usage of open science practices due to the added time and resources to mandate the change [49]. Moreover, support from editors is crucial for promoting open science practices reforms within journals. Despite overall positive attitudes towards open science, a survey conducted by Naaman et al. identified that editors see the time and effort required to implement open science practices as a major barrier for open science promotion at the journal level [49]. For example, if data citation was mandated as a condition of publication, peer reviewers may not want to take additional time to ensure articles are published in accordance with the journal's open science practices. Additionally, the current academic publishing culture favors publications, most of which report statistically positive results. [50]. Open science is a relatively new set of principles and practices which might explain why some authors have not yet integrated them into their research ecosystem. Publishers can potentially alleviate this barrier for both authors and editors by changing the default settings in popular manuscript submission systems to allow for more uniform data input in line with TOP (e.g., a mandatory field requiring input of a data availability statement) [51]. Such changes have been attempted in a study performed by Giofrè et al. that suggests that journal-specific submission guidelines could promote positive alterations in author practices [15]. This adjustment would require authors to include TOP items in the manuscript submission, requiring author adherence to open science practices and reducing the time and effort required from peer reviewers to check for usage of open science practices. Based on this discussion, a next step is to investigate how well CAIM authors adhere to open science practices encouraged or required by these CAIM journals. This type of study can allow comparison of standards set by journals with the actual implementation of open science practices by authors in the field.

Another barrier to the implementation of open science practices at a journal level stems from the frequent poor reporting of CAIM research [52]. When research is poorly reported, there is a decreased ability to draw conclusive results and make comparison with similar research studies, reducing research efficiency, validity, and replicability [52,53]. The multi-modal treatment delivery of some types of CAIM are not well substantiated by research, often limiting its focus to one or two treatment interventions [53]. For example, while acupuncture needling is researched as the primary technique employed by licensed acupuncturists, their use of other interventions (e.g., cupping and massage) that are also employed in their multi-modal treatment plans are not well researched [53]. Adapting research methods commonly used in Western biomedicine, such as clinical trials, for use in CAIM research in addition to using other research methods such as qualitative studies can improve its research basis [52,54]. Components of open science practice can support this research through the requirement of data sharing and a methodologically sound study design. Furthermore, there is low support for CAIM research from institutions, leading to its inadequate research infrastructure development and funding for research training in the field [54,55]. By training CAIM researchers in open science practices, which emphasize open data sharing and preregistration, its encouraged

use can improve the rigor of CAIM study reporting and methodology [53]. Consequently, the number of inadequately designed studies can be reduced, thereby making research more efficient [53]. With improved research and open science training, CAIM researchers may be more inclined to use open science practices in research development, which also makes journal level open science practice changes more likely to be made. For example, in the field of economics, researcher attitudes are readily accepting of strong transparency standards, which has led many high-impact economics journals to also adopt strong open science requirements [7].

In addition to journals, it is also important to consider the usage of open science practices and barriers authors face with respect to its implementation in their fields, as they are primarily affected by open science mandates in journals. A survey study conducted on author usage of open science practices found that while many authors are familiar with the idea of open science, many lack knowledge on how to implement these practices [56]. In another survey on CAIM author attitudes about open science, funding was found to be a major barrier to applying open science practices in research [57]. Both of these findings can serve as starting points for investigating how journals can better facilitate the use of open science research practices. Other fields of research have also initiated collaborative efforts between different members of the scientific community to improve open science practices. For example, during an expert meeting organized by the European Health Psychology Society to improve the use of open science best practices in health psychology, it was suggested that members of the scientific community themselves, including researchers and editors, identify and work to resolve barriers that prevent adoption of open science practices in their own field [58]. Overall, further research focused on gathering and implementing feedback from CAIM experts as part of the ongoing improvement and implementation process of open science practices may improve the use and efficacy of open science practices in CAIM.

Furthermore, increased funding needs to be directed towards CAIM research to train and incentivize these practitioners to perform research in this field. While many research disciplines face challenges with limited funding, the allocation of funding for CAIM research is notably lower compared to other healthcare sectors [59]. The safety and efficacy of CAIM therapies have been long debated and uncertainty with findings in the field significantly contribute to the lack of financing allocated to CAIM research [22]. Failure to obtain CAIM research funding in the past has often been attributed to skepticism and low perceived priority of CAIM research by grant reviewers, as well as low institutional support of CAIM research [60]. Funding applications for CAIM research have also been seen to fall short in comparison to conventional medicine due to insufficient infrastructure and fewer university level researchers showing interest in the field [22,55]. As noted by Veziari et al., as compared to advocating for linear research funding models, creating multi-faceted funding programs, though previously limited to other disciplines, may be favorable for CAIM as well [61] For example, funding research programs that require interaction between sponsors, CAIM researchers, and users, which include practitioners and patients, allows the opportunity to identify and address misconceptions regarding quality of findings within the field [52,62]. Investing in CAIM research training can also promote research quality that meets a more comparable standard to conventional medical research, and, in turn, makes for more competitive funding applications [55]. The adoption of open science practices, itself, provides an opportunity to present transparent, reproducible, and evidence-based CAIM research, which can also incentivize funding agencies to invest in CAIM research [33].

Several fields of research have seen improvements in the quality of their research articles and other metrics after implementing open science practices. Studies in the field of psychology have focused on registered reports, which is a type of registration that requires a two-stage peer review process [63]. A protocol's introduction, materials, and methods undergo a first

stage of peer review prior to the execution of the methodology, and is peer reviewed a second time once the manuscript is completed, checking for adherence to the original methodology [63]. Soderberg et al. found that psychology research articles published with registered reports have significantly improved research thoroughness and reputability, as well as include a more rigorous methodology and analysis [64]. Additionally, Obels et al. found that compared to other articles, registered reports have higher rates of data and code sharing [65]. However, studies have also found that even when data sharing is required by journals, data reporting is often inadequate for reproducibility which shows that open data alone is unable to achieve the proposed benefits of data sharing [66,67]. To draw conclusions about the impact of open science in CAIM, more research is warranted.

## Strengths and limitations

Our study has notable strengths. We sourced our subset of CAIM journals from a bibliometric analysis conducted by Ng, which was sourced from the Scopus Source List, a list of journals with established ASJC categories [37]. Furthermore, screening, data extraction, and TOP rubric assessment of the CAIM journals was conducted independently and in duplicate. All collected information was reviewed by all authors, and any discrepancies was resolved via consensus. Furthermore, comparing definitions against the TOP rubric provided a standardized reference for comparing data across different journals. The detailed and specific methodology also allows for the replicability of the study's findings.

Regarding limitations, this study only includes CAIM journals that published online, which may not reflect the open science practices of CAIM journals that only publish in print format. Additionally, only journals that publish in the English language were included, thus our findings may not be representative of CAIM journals published in other languages. Furthermore, the audit relies on the information made publicly available by the included journals. If relevant data on open science practices was not readily accessible or inconsistently reported, the study's findings may be incomplete or inaccurate. We also acknowledge that the audit may not capture the reasons behind the observed open science practices. Without additional qualitative data or information from journal editors, authors, or reviewers, it can be challenging to fully understand the motivations or barriers influencing open science practices in CAIM journals. Further, this audit represents a snapshot of open science practices at a specific point in time, and it can be expected that this information may change over time. Lastly, the TOP guidelines also do not address open science practices in regard to open access publishing models, which is a notable aspect of open science.

## Conclusions

In conclusion, it was found that CAIM journals provide minimal guidelines to encourage or require authors to adhere to open science practices. This audit serves as a starting point to understand and improve the usage of open science practices in CAIM journals. The inclusion of greater open science practices in CAIM journals may serve to enhance the usability and replicability of research published in these journals. Knowledge of how open science practices are encouraged within CAIM journals can inspire initiatives that aim to make research within this field more accessible to researchers and healthcare providers.

## Supporting information

**S1 Table. Specific text associated with each of the eight TOP guidelines for the 19 CAIM journals.**
(XLSX)

## Acknowledgments

We gratefully acknowledge Navila Asgar for her contributions and review of our study protocol.

## Author Contributions

**Conceptualization:** Jeremy Y. Ng.

**Data curation:** Jeremy Y. Ng, Brenda Lin, Tisha Parikh.

**Formal analysis:** Jeremy Y. Ng, Brenda Lin, Tisha Parikh.

**Investigation:** Jeremy Y. Ng, Brenda Lin, Tisha Parikh.

**Methodology:** Jeremy Y. Ng, Holger Cramer, David Moher.

**Project administration:** Jeremy Y. Ng.

**Resources:** Holger Cramer.

**Supervision:** Jeremy Y. Ng, Holger Cramer, David Moher.

**Writing – original draft:** Jeremy Y. Ng, Brenda Lin, Tisha Parikh.

**Writing – review & editing:** Jeremy Y. Ng, Brenda Lin, Tisha Parikh, Holger Cramer, David Moher.

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
