## [Decision Letter · Decision Letter 0]

28 Feb 2024

PONE-D-23-42283Investigating the Nature of Open Science Practices Across Complementary, Alternative, and Integrative Medicine Journals: An AuditPLOS ONE

Dear Dr. Ng,

Thank you for submitting your manuscript to PLOS ONE. After careful consideration, we feel that it has merit but does not fully meet PLOS ONE’s publication criteria as it currently stands. Therefore, we invite you to submit a revised version of the manuscript that addresses the points raised during the review process.

I agree with both reviewers about the need for revisions of the manuscript.

We look forward to receiving your revised manuscript.

Kind regards,

Diego A. Forero, MD; PhD

Academic Editor

PLOS ONE

Journal Requirements:

2. Please upload a copy of Figure 1, to which you refer in your text on page 8. If the figure is no longer to be included as part of the submission please remove all reference to it within the text.

Additional Editor Comments:

I agree with both reviewers about the need for revisions of the manuscript.

Reviewers' comments:

Reviewer's Responses to Questions

**Comments to the Author**

1. Is the manuscript technically sound, and do the data support the conclusions?

Reviewer #1: Partly

Reviewer #2: Yes

2. Has the statistical analysis been performed appropriately and rigorously? 

Reviewer #1: N/A

Reviewer #2: Yes

3. Have the authors made all data underlying the findings in their manuscript fully available?

Reviewer #1: Yes

Reviewer #2: Yes

4. Is the manuscript presented in an intelligible fashion and written in standard English?

Reviewer #1: Yes

Reviewer #2: Yes

5. Review Comments to the Author

Reviewer #1: I appreciate the author’s attention to this topic as well as the transparency in the supplemental files and also the explicit detail on the articles included. I also appreciate the specificity in the discussion that holds potential to help to advance rigor in research practices in the field. I feel it will be a valuable addition to the scholarly literature with some additional clarifications that include the following:

• It would be helpful if the authors included some additional information according to PRISMA guidelines for systematic reviews such as the flow chart showing inclusions and exclusions as well as additional information on their initial screening procedures and interrater reliability: http://www.prisma-statement.org/?AspxAutoDetectCookieSupport=1 I do note that the authors state that Figure 1 is the flow chart for inclusion/exclusion but I could not find a Figure in the PDF document for review nor the supplemental files linked. I apologize if I missed in somewhere but did try to find it and also did document searches for it.

• I am not an expert in contemporary and alternative medicines so I cannot verify if the eligibility criteria for particular journals would accurately capture this field of literature. Therefore, it would be helpful to readers if the authors could clarify how they came up with the criteria for inclusion/exclusion and whether: a) they consulted an expert from the field of CAIM to ensure they are accurately capturing the literature or b) if perhaps there are other systematic reviews conducted within the field of CAIM that utilized similar inclusion criteria for journals. Overall, I just would like more information verifying that the eligibility criteria accurately captured the literature of the particular field of CAIM. I put this note in because there have been other systematic reviews that have not adequately represented the field of expertise due to their inclusion/exclusion criteria; ultimately resulting in bias due to missing data. Hence, I think a few clarifying statements ensuring how this criteria adequately captures the breadth of literature of CAIM would be helpful in understanding and would ultimately mitigate these questions.

• Last, I feel it would be helpful to perhaps consider some points in the discussion about the potential need for gathering input from the field of CAIM to identify their needs, concerns, and supports relevant to implementing more OS practices and TOP Guidelines as part of next steps and future research. For example, with qualitative research there have been concerns and relevant points identified in recent years related to the use of OS practices and also implementation of TOP guidelines. Advancement in this area of OS practices in qualitative research has occurred due to explicit scholarly discourse on the topic that includes varied viewpoints as well as recent collaborations between OS experts and experts in the field of qualitative research. I feel this also is relevant when it comes to the needs of particular fields of disciplinary expertise. Although I would agree that we can all benefit from improving our research and practices, there will be nuances and potential impacts specific to particular disciplines and professional areas of expertise. Therefore, future studies that focus on gathering feedback from field experts as part of the ongoing improvement and implementation process of guidelines will only strengthen all fields of study, and perhaps the TOP Guidelines as well.

Thank you for the opportunity to review. I appreciate the time of the authors in examining this topic of importance.

Reviewer #2: The manuscript describes an audit conducted to investigate open science practices in complementary, alternative, and integrative medicine (CAIM) journals. It utilized established guidelines and a systematic approach to selecting journals, extracting data, and evaluating open science practices. The results indicate that CAIM journals generally provide minimal guidelines to encourage or require authors to adhere to open science practices.

The statistical analysis employed in the study was descriptive, aligning with the study's aims to examine open science practices in CAIM journals. It involved summarizing journal characteristics and TOP guideline scores using measures like mean, median, range, and standard deviation. This approach is suitable for providing a comprehensive overview of open science practices across this journals.

All relevant data are included in this manuscript or posted on Open Science Framework at the link https://doi.org/10.17605/OSF.IO/S7G6P

Recommendations:

Highlighting in the Strengths section of this research the explicit improvement of open science practices in these journals could benefit both researchers and healthcare professionals, as well as patients who use complementary and integrative medicine treatments.

Explaining in more detail the possible factors that contribute to the low scores of open science practices in complementary and integrative medicine journals, such as barriers they may face: for example, social, cultural, and economic barriers.

6. PLOS authors have the option to publish the peer review history of their article (what does this mean?). If published, this will include your full peer review and any attached files.

Reviewer #1: **Yes: **Sondra Stegenga

Reviewer #2: No

---

## [Author Response · Author response to Decision Letter 0]

18 Mar 2024

Response to Reviewers

Dear Editor:

Thank you for your email. I am pleased to hear that you believe that our manuscript will have the potential to be published in PLOS ONE following requested revisions.

As per your request, please find our response to the peer-reviewers’ comments with every change outlined point by point below:

Journal Requirements:

2. Please upload a copy of Figure 1, to which you refer in your text on page 8. If the figure is no longer to be included as part of the submission please remove all reference to it within the text.

Additional Editor Comments:

I agree with both reviewers about the need for revisions of the manuscript.

• We kindly thank the editor for providing their feedback on our manuscript.

Reviewers' comments:

Reviewer's Responses to Questions

Comments to the Author

1. Is the manuscript technically sound, and do the data support the conclusions?

Reviewer #1: Partly

Reviewer #2: Yes

2. Has the statistical analysis been performed appropriately and rigorously?

Reviewer #1: N/A

Reviewer #2: Yes

3. Have the authors made all data underlying the findings in their manuscript fully available?

Reviewer #1: Yes

Reviewer #2: Yes

4. Is the manuscript presented in an intelligible fashion and written in standard English?

Reviewer #1: Yes

Reviewer #2: Yes

5. Review Comments to the Author

Reviewer #1: I appreciate the author’s attention to this topic as well as the transparency in the supplemental files and also the explicit detail on the articles included. I also appreciate the specificity in the discussion that holds potential to help to advance rigor in research practices in the field. I feel it will be a valuable addition to the scholarly literature with some additional clarifications that include the following:

• We kindly thank this reviewer for providing their feedback on our manuscript.

• It would be helpful if the authors included some additional information according to PRISMA guidelines for systematic reviews such as the flow chart showing inclusions and exclusions as well as additional information on their initial screening procedures and interrater reliability: http://www.prisma-statement.org/?AspxAutoDetectCookieSupport=1 I do note that the authors state that Figure 1 is the flow chart for inclusion/exclusion but I could not find a Figure in the PDF document for review nor the supplemental files linked. I apologize if I missed in somewhere but did try to find it and also did document searches for it.

• We would like to clarify that we conducted an audit rather than a systematic review, meaning PRISMA guidelines would not apply. However, we understand the importance of clearly describing our eligibility criteria and created that flow chart to show our process. Unfortunately, we did not include it in the initial submission, and we apologise for this confusion. We have now added the missing figure as a file to the revised submission. 

• I am not an expert in contemporary and alternative medicines so I cannot verify if the eligibility criteria for particular journals would accurately capture this field of literature. Therefore, it would be helpful to readers if the authors could clarify how they came up with the criteria for inclusion/exclusion and whether: a) they consulted an expert from the field of CAIM to ensure they are accurately capturing the literature or b) if perhaps there are other systematic reviews conducted within the field of CAIM that utilized similar inclusion criteria for journals. Overall, I just would like more information verifying that the eligibility criteria accurately captured the literature of the particular field of CAIM. I put this note in because there have been other systematic reviews that have not adequately represented the field of expertise due to their inclusion/exclusion criteria; ultimately resulting in bias due to missing data. Hence, I think a few clarifying statements ensuring how this criteria adequately captures the breadth of literature of CAIM would be helpful in understanding and would ultimately mitigate these questions.

• We would like to clarify that we conducted an audit rather than a systematic review. 

• We initiated the selection process using the list of CAIM journals from Ng’s bibliometric analysis since it was derived from Scopus, the largest and most relevant database for our purposes. We wanted our audit to focus only on general CAIM journals rather than journals focused on a specific CAIM therapy (ex. homeopathy), as any CAIM topic can go into a general CAIM journal but not vice versa with general CAIM research nearly never being published in those specific journals. As a result, we modified Ng’s list to only include journals with “complementary,” “alternative,” or “integrative” in their titles. We note that Dr. Holger Cramer and Dr. Jeremy Ng are experts on the subject area of CAIM, and they were involved in creating and approved of final the eligibility criteria used in this audit. We understand our reasoning may have been unclear in the Methods section and have added this clarification in the revised submission. We also appreciate your feedback regarding the breath of literature covered and, in the discussion, we proposed to audit specific CAIM journals in future research endeavours. 

• Last, I feel it would be helpful to perhaps consider some points in the discussion about the potential need for gathering input from the field of CAIM to identify their needs, concerns, and supports relevant to implementing more OS practices and TOP Guidelines as part of next steps and future research. For example, with qualitative research there have been concerns and relevant points identified in recent years related to the use of OS practices and also implementation of TOP guidelines. Advancement in this area of OS practices in qualitative research has occurred due to explicit scholarly discourse on the topic that includes varied viewpoints as well as recent collaborations between OS experts and experts in the field of qualitative research. I feel this also is relevant when it comes to the needs of particular fields of disciplinary expertise. Although I would agree that we can all benefit from improving our research and practices, there will be nuances and potential impacts specific to particular disciplines and professional areas of expertise. Therefore, future studies that focus on gathering feedback from field experts as part of the ongoing improvement and implementation process of guidelines will only strengthen all fields of study, and perhaps the TOP Guidelines as well.

• We thank you for this suggestion. We, too, find this future direction relevant and meaningful as you have explained. We have added in a short paragraph summarizing this point with appropriate sources cited where needed.

Thank you for the opportunity to review. I appreciate the time of the authors in examining this topic of importance.

Reviewer #2: The manuscript describes an audit conducted to investigate open science practices in complementary, alternative, and integrative medicine (CAIM) journals. It utilized established guidelines and a systematic approach to selecting journals, extracting data, and evaluating open science practices. The results indicate that CAIM journals generally provide minimal guidelines to encourage or require authors to adhere to open science practices.

The statistical analysis employed in the study was descriptive, aligning with the study's aims to examine open science practices in CAIM journals. It involved summarizing journal characteristics and TOP guideline scores using measures like mean, median, range, and standard deviation. This approach is suitable for providing a comprehensive overview of open science practices across this journals.

All relevant data are included in this manuscript or posted on Open Science Framework at the link https://doi.org/10.17605/OSF.IO/S7G6P

• We kindly thank this reviewer for providing their feedback on our manuscript.

Recommendations:

Highlighting in the Strengths section of this research the explicit improvement of open science practices in these journals could benefit both researchers and healthcare professionals, as well as patients who use complementary and integrative medicine treatments.

• We appreciate your recommendation. We believe that the Strengths section should focus on methodological strengths, however we do touch on this strength in the second paragraph of the discussion section. Based on your recommendation, we have since added more explicit information on how improvement of open science practices can benefit these individuals. 

Explaining in more detail the possible factors that contribute to the low scores of open science practices in complementary and integrative medicine journals, such as barriers they may face: for example, social, cultural, and economic barriers.

• We thank you for this suggestion. We have added more information about cultural barriers that CAIM journals may face with respect to open science implementation. 

6. PLOS authors have the option to publish the peer review history of their article (what does this mean?). If published, this will include your full peer review and any attached files.

Do you want your identity to be public for this peer review? For information about this choice, including consent withdrawal, please see our Privacy Policy.

Reviewer #1: Yes: Sondra Stegenga

Reviewer #2: No

Should your editorial office require any further edits following my most recently submitted submission, please do not hesitate to inform me and we will make these changes as soon as possible. Thank you for your consideration.

Yours sincerely,

Dr. Jeremy Y. Ng, MSc, PhD

Postdoctoral Fellow, Centre for Journalology, Ottawa Methods Centre, Ottawa Hospital Research Institute, Ottawa, Canada

Research Associate, Institute of General Practice and Interprofessional Care, University Hospital Tübingen, Tübingen, Germany & Robert Bosch Center for Integrative Medicine and Health, Bosch Health Campus, Stuttgart, Germany

---

## [Decision Letter · Decision Letter 1]

10 Apr 2024

Investigating the Nature of Open Science Practices Across Complementary, Alternative, and Integrative Medicine Journals: An Audit

PONE-D-23-42283R1

Dear Dr. Ng,

We’re pleased to inform you that your manuscript has been judged scientifically suitable for publication and will be formally accepted for publication once it meets all outstanding technical requirements.

Kind regards,

Diego A. Forero, MD; PhD

Academic Editor

PLOS ONE

Additional Editor Comments (optional):

Reviewers' comments:

Reviewer's Responses to Questions

**Comments to the Author**

1. If the authors have adequately addressed your comments raised in a previous round of review and you feel that this manuscript is now acceptable for publication, you may indicate that here to bypass the “Comments to the Author” section, enter your conflict of interest statement in the “Confidential to Editor” section, and submit your "Accept" recommendation.

Reviewer #1: All comments have been addressed

Reviewer #2: All comments have been addressed

2. Is the manuscript technically sound, and do the data support the conclusions?

Reviewer #1: Yes

Reviewer #2: Yes

3. Has the statistical analysis been performed appropriately and rigorously? 

Reviewer #1: Yes

Reviewer #2: Yes

4. Have the authors made all data underlying the findings in their manuscript fully available?

Reviewer #1: Yes

Reviewer #2: Yes

5. Is the manuscript presented in an intelligible fashion and written in standard English?

Reviewer #1: Yes

Reviewer #2: Yes

6. Review Comments to the Author

Reviewer #1: I continue to feel this is a manuscript of high relevance . I feel that feedback has been addressed both within the manuscript and in responses. I appreciate the detailed revisions and additions. Thank you for your time in this needed work.

Reviewer #2: All comments were replied to and explained by the author, and they were incorporated into the manuscript

7. PLOS authors have the option to publish the peer review history of their article (what does this mean?). If published, this will include your full peer review and any attached files.

Reviewer #1: **Yes: **Sondra Stegenga

Reviewer #2: No
